# A Review of Published Literature Regarding Health Issues of Coastal Communities in Sabah, Malaysia

**DOI:** 10.3390/ijerph17051533

**Published:** 2020-02-27

**Authors:** Amirah Azzeri, Goh Hong Ching, Hafiz Jaafar, Mohd Iqbal Mohd Noor, Nurain Amirah Razi, Amy Yee-Hui Then, Julia Suhaimi, Fatimah Kari, Maznah Dahlui

**Affiliations:** 1Department of Social and Preventive Medicine, Faculty of Medicine, University of Malaya, Kuala Lumpur 50603, Malaysia; fazilamazira@gmail.com (A.A.); drhafizjaafar@gmail.com (H.J.); 2Department of Urban and Regional Planning, Faculty of Built Environment, University of Malaya, Kuala Lumpur 50603, Malaysia; gohhc@um.edu.my (G.H.C.);; 3Centre for Sustainable Urban Planning & Real Estate (SUPRE), Faculty of Built Environment, University of Malaya, Kuala Lumpur 50603, Malaysia; 4Department of Primary Care, Faculty of Medicine and Health Sciences, Universiti Sains Islam Malaysia, Kuala Lumpur 56100, Malaysia; amirahrazi@usim.edu.my; 5Institute of Biological Science, Faculty of Science, University of Malaya, Kuala Lumpur 50603, Malaysia; amy_then@um.edu.my; 6Department of Primary Care Medicine, Faculty of Medicine, University of Malaya, Kuala Lumpur 50603, Malaysia; JULIA.SUHAIMI@ummc.edu.my; 7Department of Economics, Faculty of Economics and Administration, University of Malaya, Kuala Lumpur 50603, Malaysia; fatimah_kari@um.edu.my; 8Centre for Population Health (CePH), Faculty of Medicine, University of Malaya, Kuala Lumpur 50603, Malaysia

**Keywords:** health status, coastal, Sabah

## Abstract

Several of the coastal zones in Sabah, Malaysia, are isolated and inaccessible. This study aimed to review the published literature on the health status of the coastal communities in Sabah. The following four main health issues were found: (i) malaria, (ii) tuberculosis (TB), (iii) seafood poisoning, and (iv) antenatal problems. Factors associated with the risk of acquiring malarial infection in the studied coastal area were advanced age, male sex, farming as an occupation, history of travel outside the village, and rainy seasons. TB infection was primarily observed in adult men. Seafood poisoning was significantly common in Sabah. Studies have reported that tetrodotoxin and paralytic shellfish poisoning were commonly reported (30–60 cases annually). Several pregnant women in the coastal community had insufficient knowledge of the national antenatal care programme. Nonetheless, 99% of them received antenatal care at public healthcare facilities with 92% of them undergoing safe delivery. Nevertheless, a majority of the pregnant women had iodine deficiency due to low iodised salt intake. Findings from this review highlighted that the coastal communities in Sabah are experiencing significant health problems. Specific attention is required to significantly enhance the health and well-being of the individuals living in the coastal communities in Sabah.

## 1. Introduction

A coastal community is described as a coastal settlement where people live on the thin strip of land or on the water along the boundaries between the sea and the land including seaside towns and ports [1]. The communities are facing several health issues due to environmental and social factors related to the area. Environmental factors such as climate changes, habitat modifications, air and water pollution, and geographical remoteness of the coastal areas lead to serious health problems [2]. Social factors such as high poverty rate, unstable income, and limited access to health and education are associated with negative health behaviours [1,3].

Globally, the term coastal area or coastal zone is inconsistently described considering the significant variation in topographical demarcation across countries [1,4,5,6]. A previous study in Malaysia described that the width of the coastal area was between 1 and 5 km from the shoreline [6]. Considering that Malaysia is bounded by the South China Sea, the Sulu Sea, and the Strait of Malacca, several people are living in the coastal area and are dependent on marine and coastal ecosystems for food, employment, and general well-being. It is estimated that Malaysia is covered by 4.43 million hectares of the total coastal area, which is equivalent to 13% of the land in the country [5]. Sabah is one of the states in Malaysia with a high number of coastal zones, approximately constituting greater than 10% of the total region [6].

The coastal areas in Sabah are possibly more different compared to other coastal areas in Peninsular Malaysia and Sarawak. Several of the coastal zones in Sabah are isolated and inaccessible [7]. Although public healthcare facilities are available in the coastal areas of Sabah, the facilities provided are limited and might not reach those who are in need due to several reasons such as poverty, poor health awareness, and insufficient infrastructures used to access the healthcare services [8]. Moreover, the coastal areas in Sabah are experiencing the highest coastal erosion, which affects the livelihood and health of the community [9]. Sabah has the highest poverty rate in Malaysia, and several coastal areas in Sabah such as Kudat, Kota Marudu, Pitas, Kunak, and Semporna are reported to have poverty rates nearly as high as 50% [10]. Furthermore, the number of stateless and undocumented individuals is also high in Sabah’s coastal areas [11]. Considering these factors, the coastal communities are exposed to several health problems, which could possibly result in poor health status.

To date, a systematic review aimed to determine the health status of the coastal communities in Sabah has not been conducted yet. This systematic review is one of the components of a large-scale research project that determines the health status, health problems, health gaps, health needs, health utilisation, and quality of life of the individuals living in the coastal communities in Sabah. Generally, a large-scale research project was planned to understand the health problems in the coastal area studied and to recommend appropriate intervention programmes to improve the overall health status of coastal communities in the future. Various sub-studies were planned to achieve the general objectives of the large-scale project. For example, in the first part of this large-scale research project, a comprehensive systematic review of the available literature on the health issues of the coastal communities in Sabah was conducted to obtain an overview regarding the health problems and health gaps among the coastal communities. The second part will include health surveys on the general health, healthcare utilisation, and quality of life of individuals living in the coastal communities. The third part of this large-scale project will include in-depth interviews and focus group discussions with relevant authorities, particularly those from the Ministry of Health who are dealing with the coastal communities in Sabah. Simultaneously, data collection from the national registries and annual report will be collected accordingly. This systematic review aimed to review the published and available literature on the health status of the coastal communities in Sabah. This information is significantly beneficial in understanding the health issues, health gaps, and health needs of individuals living in coastal areas.

## 2. Materials and Methods

### 2.1. Literature Search

This review focuses on identifying published articles that have reported the health status of coastal communities in Sabah. For the purpose of this review, the health status is a generic term that refers to the health of the coastal communities in Sabah, which was assessed either through objective or subjective measures.

A structured and systematic electronic search strategy was conducted using the traditional search and citation forward and backward tracking using the following three computerised databases: PubMed, Scopus, and Web of Science (WoS). The literature search was conducted using specific keywords and identified Medical Subject Heading (MeSH) terms for PubMed in May 2019. The search strategy and keywords used are shown in Table 1.

Several inclusion and exclusion criteria were applied in the literature search. The inclusion criteria were as follows: (1) studies that were published within a 10-year period from 2009 to 2019, (2) full article journals, (3) studies that elicited the health status/problem of the coastal communities in Sabah, and (4) studies that were published in English language. Initially, the review intended to capture all literature without a specific time frame. However, the literature prior to 2009 reported diseases that occurred occasionally such as cholera and filariasis and malaria infection due to *Plasmodium falciparum* (*P. falciparum*) and *Plasmodium vivax* (*P. vivax*), which were currently well controlled and almost eliminated nationwide. Some of the articles on *P. falciparum* and *P. vivax* have been published since 1947.

Studies that described the non-human health status such as studies on pathogens, animals, or environmental conditions in the coastal areas, studies that reported the health status of the non-coastal communities in Malaysia, and studies that reported the cultural or ritual activities of people in Sabah, which were not health-related, were excluded.

### 2.2. Data Extraction

Four authors independently extracted the information from the relevant articles. Information on the study design, measures of health status, and reported outcomes were summarised in a standardised evidence table. Data were compared, and disagreements were resolved by discussion to reach a consensus. Further analysis and quality assessment of the selected articles were performed by a majority of the authors.

### 2.3. Quality Assessment

Studies were evaluated for methodological quality using the standard quality assessment criteria for evaluating primary research papers from various field’s checklists [12]. The checklist has been considered a reliable tool for use in a systematic review with multiple research topics and various study designs.

Quantitative studies were evaluated based on 14 criteria. The 14 criteria included the following: (1) study objectives; (2) study designs; (3) participant recruitments; (4) participant characteristics; (5) selection process of the study participants; (6) blinding process, if required for the investigators; (7) blinding process, if required for the study participants; (8) study outcomes; (9) sample size calculations; (10) findings analyses; (11) variance estimations for the main results; (12) control of the confounders; (13) reported sufficient results; and (14) conclusion. The aforementioned 14 criteria were scored depending on the degree to which the specific criteria were met (‘yes’ = 2, ‘partial’ = 1, ‘no’ = 0). Items not applicable to a particular study design were marked ‘n/a’ and were excluded from the calculation of the summary score. A summary score was calculated for each paper by summing the total score obtained across the relevant items and dividing it by the total possible score and subsequently timed by 100 to obtain the percentage [12].

Regarding the qualitative study, 10 criteria were used to evaluate the quality of the study, which included the following: (1) study objective, (2) study design, (3) study context, (4) connection to a theoretical framework, (5) sampling strategy, (6) data collection methods, (7) findings analysis, (8) use of verification procedure to establish credibility, (9) conclusion, and (10) account reflexivity. Similarly, the score was obtained depending on the degree to which the specific criteria were met (‘yes’ = 2, ‘partial’ = 1, ‘no’ = 0). The summary score was subsequently calculated by summing the total score obtained across the relevant items and dividing it by the total possible score and subsequently timed by 100 to obtain the percentage [12]. A score was calculated for each study, which could subsequently be classified into good-quality category if the score obtained is 75% and above.

### 2.4. Flow of Study Selection

The flow of the study selection is shown in Figure 1. The potentially relevant articles were retrieved using three databases (PubMed, Scopus, and WoS). All articles were examined for duplications, and duplicate articles from the databases were removed. Title and abstract screenings were conducted in the articles. Articles that were found to be relevant and published in the specified time frame were included. Additionally, several articles were identified from a hand search of bibliographic references of the shortlisted articles. Subsequently, a full-text screening was conducted on the remaining articles. Articles that did not meet the inclusion and exclusion criteria described were further excluded.

### 2.5. Ethics

This study was conducted in accordance with the conditions of ethics clearance obtained from the University of Malaya Research Committee (Ref. No.: UM. TNC2/UMREC—408) and University of Exeter College of Medicine and Health (Ref. No.: 19/06/214) and research permits obtained from Sabah Parks (Ref. No.: TTS/IP/100-6/2 Jld. 8 [78]) and Sabah Biodiversity Centre (Ref. No.: JKM/MBS.1000-2/2 JLD.8).

## 3. Results

### 3.1. General Characteristics of the Included Articles for the Review

A total of 11 articles were finally considered and included in this review. Overall, the most recent articles were published in 2017 [13,14,15,16,17], and the oldest article was published in 2010 [18]. All of the included articles were descriptive studies. All articles were quantitative studies except for one article, which was a qualitative study [18]. Three studies were conducted specifically at Kudat, two studies were conducted at Kudat and Kota Marudu, one study was conducted at Kota Kinabalu, and the remaining five studies were conducted at the state level of Sabah, which included several coastal areas such as Kota Belud, Putatan, Pitas, Tuaran, Membakut, and Keningau.

Five out of the 11 articles reported on malaria infection [13,19,20,21,22], two of the articles reported on seafood poisoning [15,16], and two articles reported on tuberculosis (TB) disease [18,23]. A study assessed the antenatal care practice and pregnancy outcome [17], and another study measured iodine status among pregnant women [14]. Overall, the 11 appraised articles had good-quality evidence ranging from 77% to 100% based on the assessment conducted. The Appendix A present the description of the 11 articles reviewed and the quality scores based on the assessment tool.

### 3.2. Health Status

Health status was reported in several aspects in this review and was categorised into the following four categories: (i) incidences, distributions, disease characteristics, and factors associated with malaria infection; (ii) demographic characteristics, knowledge, and perceptions of TB patients; (iii) incidence of seafood poisoning; and (iv) antenatal problems experienced by pregnant women in the coastal areas of Sabah.

#### 3.2.1. Incidence, Distributions, Disease Characteristics, and Factors Associated with Malaria Infection

The incidence of malaria infection from 2009 to 2011 was estimated to be 2.6/1000 people/year [19], and the most common species found was *Plasmodium knowlesi* (*P. knowlesi*), which ranged from 76% to 78% of the total cases in Sabah [19,20] with an increased notifications from 703 in 2011 to 815 in 2012 and 996 in 2013. It was found that there were several factors associated with the risk of acquiring *P. knowlesi* infection in the coastal communities in Sabah. The factors can be divided into socio-demographic profiles, human activities, housing environments, and seasonal variations [13,19,20,21,22].

Barber et al. (2012) reported that the *P. knowlesi* infection affected various adult age groups. The median age of patients with *P. knowlesi* infection was significantly higher than that of patients diagnosed with other types of *Plasmodium* species infections. The adjusted odds ratio for *P. knowlesi* infection among adults aged 15 years and above was 4.16, with men reported to have higher odds ratio for *P. knowlesi* infection than women, mainly due to occupational exposure to forest areas [13,20]. Individuals with glucose-6-phosphate dehydrogenase (G6PD) deficiency have higher risk for *P. knowlesi* infection but not with other *Plasmodium* species infections than individuals with no G6PD deficiency.

Occupations, outdoor activities, and housing environments were also more significantly associated with *P. knowlesi* infection compared to other *Plasmodium* species infections [13]. Exposure to the forests increased the likelihood of acquiring *P. knowlesi* infection considering that the macaques (the host) are primarily infected with the species, and humans can be infected from the host by spending a significant amount of time in the farms or forested areas. Additionally, outdoor activities such as sleeping outside the house and history of travel outside the village in the past 4 weeks were independently associated with the increased risk of infection [13,20]. Being aware of the presence of monkeys, having open eaves or gaps in walls, and having long grass around the house also increased the risk for *P. knowlesi* infection. By contrast, frequent insecticide spraying of household walls was associated with decreased risk of *P. knowlesi* infection [13].

*P. knowlesi* infection was associated with seasonal variations [19,20,21], precisely during rainy season. The peak incidence of *P. knowlesi* malaria infection was observed during the months with higher number of rainfalls. Besides the environmental factors, family clustering, which was defined as patients in the same household presenting with malaria infection within a 4-week period, was reported, suggesting that indoor transmission of the disease was possible from actively infected patient to another household [19].

One study investigated the demographic, clinical, and laboratory features of *P. knowlesi* infection among children [22]. The *P. knowlesi* infection comprised 59% of total malaria paediatric cases. The study found that the duration of fever was shorter for *P. knowlesi* infection compared to other *Plasmodium* species infections. Anaemia was common with *P. knowlesi* infection with greater than 80% of the children having a haemoglobin level of less than 11.0 g/dL at initial presentation. Furthermore, all children admitted with *P. knowlesi* malaria had thrombocytopaenia with a platelet count of less than 150,000 μL at admission. Nevertheless, the response to treatment with antimalarial drugs was superior for *P. knowlesi* infection with a median parasite clearance time of 2 days compared to greater than 10 days in other *Plasmodium* species infections.

#### 3.2.2. Demographic Characteristics, Knowledge, and Perceptions of Tuberculosis Patients

Other than malaria infection, TB cases are also high in Sabah. TB infection was primarily observed in men more than in women with a median (range) age of 30 (17–53) years. It was reported that several of the patients were diagnosed at the later stage of the disease with haemoptysis in nearly half of the patients (47%), high smear grade in one-third of the study population, and cavitary findings from radiological investigation observed in 64% of the diagnosed patients. The median reported duration of the symptoms prior to the initiation of treatment was approximately 8 weeks. More than half of the patients were current or ex-smokers prior to TB treatment with diabetes mellitus as the most common comorbidity followed by hypertension and asthma. Otherwise, the human immunodeficiency virus (HIV) co-infection rate was low among the TB patients in the coastal areas of Sabah [23].

Knowledge and perception of TB disease, patients’ experiences on healthcare services related to TB disease, and impacts of the disease were also reported in one study [18]. A total of 96% of the patients had insufficient knowledge regarding the causes and risk factors for TB disease. Despite the patients’ insufficient knowledge, 98% of the study population believed that TB was a contagious disease. Majority of the patients sought medical attention only when the symptoms worsened such as when they had haemoptysis, difficulty in breathing, or lethargy. Nevertheless, the patients were dependent on modern medicines prescribed by certified medical personnel, and 91% of the patients were satisfied with the healthcare services they received for TB management [18].

The TB disease had significant impacts on the affected patients. Several of the male patients felt unhealthy with poor physical performance even after treatment completion. Stigmatisation following a diagnosis of TB affected their family relationship, marital relationship, and business activities as TB was considered as a severe and untreatable disease by the community. Daily life routines of positive TB patients were significantly affected because they were isolated and asked to use separate utensils. Marital relationship was severely affected as the husbands and wives diagnosed with TB were asked to practise new sleeping arrangements to prevent the transmission of the disease. Several patients reported that they received poor support system following the disease, which generally affected their socioeconomic status and income and their psychosocial well-being [18].

#### 3.2.3. Incidence of Seafood Poisoning

Seafood poisoning is also common in the coastal communities. A case-series study was conducted to investigate tetrodotoxin poisoning following the consumption of horseshoe crab (*Carcinoscorpius rotundicauda*) in one of the coastal areas in Sabah [15]. It was found that 30 tetrodotoxin cases were reported from June to August 2011 with children aged less than 10 years as the most affected age group. Of all reported patients, six were confirmed to experience poisoning, and three of the six patients died due to tetrodotoxin poisoning. The three patients who died had respiratory paralysis and cardiovascular collapse within 24 h of horseshoe crab consumption. Another 24 probable cases of poisoning were presented with clinical manifestations of an early stage of poisoning such as circumoral and lingual numbness, nausea, vomiting, and upper and lower limb numbness.

A paralytic shellfish poisoning (PSP) or saxitoxin poisoning was also reported in the coastal communities in Sabah [16]. A total of 58 poisoning cases with four deaths were reported in Kota Kinabalu from 1 January to 6 June 2013. The poisoning was found to be associated with the consumption of contaminated shellfish collected from Sepangar Bay or Kuala Penyu Bay or bought from nearby markets. The majority of the 58 patients were women. The most common symptoms manifested were circumoral, lingual, and neck numbness, which occurred less than an hour after the consumption of the contaminated shellfish. Severe symptoms such as breathing difficulty and faintness were presented by less than 10% of the patients.

#### 3.2.4. Antenatal Problems Experienced by Pregnant Women in the Coastal Areas of Sabah

Two studies included in this review described the health status of pregnant women in the coastal communities in Sabah. A study by Aye et al. (2017) assessed the antenatal care-related knowledge, antenatal practices, and outcomes of pregnancy. The authors found that 53% of the study population had insufficient knowledge regarding antenatal care. Several of the patients had insufficient knowledge about the numbers of antenatal visits required, the importance of anti-tetanus toxoid vaccination during pregnancy, and the birth spacing interval. Despite their insufficient knowledge regarding antenatal care, 99% of the patients received antenatal care at public healthcare facilities, and the outcomes of the pregnancies were good with 92% of the study population undergoing safe delivery without any intrapartum and postnatal complications [17].

Another study found that the majority of the pregnant women had iodine deficiency disorder (IDD) despite the mandatory universal salt iodisation programme conducted in Sabah since 2000 [14]. The study reported that only 1% of the patients (*n* = 5/524) presented with thyroid gland enlargement, indicating IDD with 4 of them having grade 1 goitre and one having grade 2 goitre. The median urinary iodine concentration (UIC) was lower (105 μg/L) than the reference standard (greater than 150 μg/L). More than half (60.5%) of the women had a UIC <150 μg/L (insufficient iodine), 22.8% had a UIC of 150–249 μg/L (adequate iodine), and 16.6% had a UIC ≥250 μg/L (more than adequate or excessive iodine). It was found that pregnant women in the coastal areas such as in Kudat experienced iodine insufficiency due to low iodised salt intake as a result of inadequate access to iodised salt specifically in remote areas, poor handling and use of iodised salt, and patients’ health concern for high blood pressure caused by iodised salt consumption, limiting their salt consumption during pregnancy.

## 4. Discussion

This review found that the main health problem reported in several coastal areas in Sabah was malaria infection. Nationally, Malaysia has successfully achieved malaria elimination target, which was measured by a significant reduction in the incidence of *P. falciparum* and *P. vivax* infections in Peninsular Malaysia. The previous projection estimated that Malaysia is capable of achieving a total elimination of the infection by 2020 as targeted by the World Health Organisation (WHO) [24]. Nevertheless, despite the implementation of various elimination programmes [25,26], malaria is still significantly prevalent in the coastal areas of Sabah considering the difference among *Plasmodium* species, with *P. knowlesi* being more predominant compared to *P. falciparum* and *P. vivax* [20]. The differences in *Plasmodium* species resulted in significant challenges for malaria control programmes due to the high diversity and the difference in behaviours of the parasites, vectors, and hosts of malaria infection.

The recent increase in *P. knowlesi* infection in Sabah caused a significant challenge to completely eliminate the disease. *P. knowlesi* accounted for nearly 70% of total malaria cases in Sabah, and this species is associated with more severe complications to the central nervous system of humans even with low parasitaemia densities compared to the other species of *Plasmodium*. In *P. knowlesi* infection, early diagnosis is significantly important to prevent the clinically adverse effects of this infection. Nevertheless, the diagnosis of the infection requires a comprehensive examination (such as polymerase chain reaction test), which is not available in several remote areas, resulting in the delayed diagnosis and treatment at the advanced stage of the disease in individuals residing in the rural and the coastal areas in Sabah [24].

*P. knowlesi* infection is a disease that is primarily observed in long-tailed macaques [19] throughout Sabah and Sarawak [27]. Recently, the species has been transmitted to humans through the mosquitoes as the vectors. *P. knowlesi* was found to be the most common cause of human malaria in Kudat areas [13,19]. Kudat is a coastal and rural area in Sabah that has substantial deforestation activities. The associated environmental and population changes as a result of the destruction, clear-cutting, conversion, and removal of forested areas have been hypothesised as the main drivers of the recent emergence of *P. knowlesi* infection [21]. The deforestation activities lead to the exposure of humans to the infected macaques, the host, and *Anopheles balabacensis* mosquitoes (the vector of *P. knowlesi* infection). Previously, the main potential exposure for the infection to human is only through forest exposure, as macaques are widely observed throughout Kudat District [19]. Nevertheless, a cross-transmission of the infection to humans due to the alteration of macaques’ habitat resulted in the high incidence of *P. knowlesi* species [21].

Additionally, considering that the *A. balabacensis* mosquitoes are capable of surviving at lower densities of forested areas, there is an on-going transmission of zoonotic infection from macaques to humans through the vector [19]. *A. balabacensis* mosquitoes are highly competent and resilient with high vectorial capacity and life expectancy [28]. The vector has exclusive peak biting time, which is during the early part of the evening before dusk, when the individuals living in the community rarely use protections for mosquito bites. In contrary, other *Anopheles*’ biting times are during the late evening with the peak of biting activity at midnight and early hours of the morning.

Generally, health promotion and education on preventive measurements against the vector through the use of personal protective equipment, insecticide-treated nets, and indoor residual spraying should be strengthened and used for specific vectors and hosts related to the infection. As the country is aiming towards total malaria elimination, malaria control measures that specifically target *P. knowlesi* infection is an essential principle for an integrated vector management in Sabah, particularly at the coastal areas [28,29].

Besides malaria infection, TB is a major problem in Sabah. Generally, Malaysia is classified as a country with an intermediate burden with a notification rate of less than 100 per 100,000 population [30]. The National Tuberculosis Program in Malaysia has included several control programmes focusing on early case detection among the high-risk population, improvement in quality laboratory services, training of healthcare workers, development of comprehensive surveillance, standard guidelines, and strengthening collaboration with other public and private agencies [30].

This review found that the majority of TB patients presented to clinical care during the advanced stage of the disease and their knowledge regarding the disease is insufficient. Several patients perceived that the disease resulted in significant physical, psychological, emotional, and financial implications. In 2015, Sabah contributed to the highest number of TB cases, and immigrants were the main contributing factors of the new cases in Sabah [30,31]. This was inconsistent with the TB cases in other states of Malaysia, which was due to the presence of intravenous drug users and commonly occurring among HIV patients [31]. In Sabah particularly, the main source of the disease is pulmonary TB, while non-pulmonary TB through food and non-pasteurised milk is considered not a problem. Most of the TB patients in Sabah contracted the disease through airborne transmission from infected immigrants or foreigners. A national report on TB shows that HIV and hepatitis C were not the contributing factors for TB in Sabah, and the most important factor is the infected immigrants [32].

Illegal immigration was particularly of concern in the borders and the coastline boundaries in Sabah due to the immigrants’ easy access to the state through the water. Appropriate legislation on foreign and stateless people, in addition to the comprehensive support from military units and cooperation from local governments to control the excessive influx of immigrants in Sabah, is important to totally eliminate TB [33]. The presence of a substantial number of illegal migrants in Sabah resulted in the consistent prevalence of TB disease because the infected immigrants are usually highly mobile within the country and across the borders and act as important reservoirs for disease transmission [31]. Additionally, they usually do not undergo complete health screening before entering the country and are frequently non-compliant to therapy due to their job commitment. The majority of them are labourers; therefore, a long and complicated course of treatment prevented the compliance and lowered the treatment completion rate [31], which further complicates the TB problem in Sabah. In Malaysia, the current policy to deport foreigners with active TB within a month resulted in poorer treatment success rates [34]. To avoid this problem, a national strategy to re-strategize the distribution of anti-TB treatment in Sabah is required, and treatment monitoring is important to minimise the burden and prevent the spread of TB infection in the state [35,36,37].

In addition to the two infectious diseases described earlier, several of the beaches in Malaysia are facing severe water pollution due to industrialisation, uncontrolled waste management, and manufacturing activities, placing coastal communities at risk for food poisoning due to contaminated seafood ingestion [38]. Tetrodotoxin poisoning is significantly a common cause of food poisoning along the coasts of Asia. Besides Sabah, several episodes of tetrodotoxin poisoning were recorded in the other coastal areas in Malaysia such as in Johor, Terengganu, and Sarawak [39].

PSP is also common in the coastal areas of Malaysia. Based on the national report, more than a hundred of cases of PSP occur annually in Sabah, Sarawak, Malacca, and Penang states [40,41]. PSP cases are predominantly associated with harmful algal bloom, which is a natural phenomenon where the density of microalgal species in both marine and freshwater environments increases due to physiological alteration and rapid growth of the species, resulting in the accumulation of algal-origin toxins in shellfish [40]. Coastal water pollution and systematic increase in seawater temperature are considered as the possible contributing factors to this phenomenon.

Control measures to prevent food poisoning related to seafood consumption are essential. Incomprehensive law enforcement and insufficient monitoring development activities such as industrialisation and urbanisation that lead to eutrophication of the coastal waters are the major contributors for the incidence of poisoning particularly at the remote coastal areas in Sabah that are highly polluted such as in Kudat [42]. Additionally, legislation of food acts, national surveillance, public awareness programmes, and the development of national guidelines on the appropriate preparation of marine products for consumption are essential to control the disease, resulting in an improved well-being of the coastal communities [39,40,41]. Preparing and cooking of affected seafood products require a specific technique, and most individuals have insufficient knowledge regarding this technique. Individuals living in coastal communities should be educated on the potential risks and the clinical manifestations related to the ingestion of contaminated seafood products and the importance of early medical attention.

The coastal communities in Sabah are also frequently associated with poverty and low educational background. Hence, they experience difficulty in understanding and using the provided information to promote and maintain an improved health [43] despite the healthcare services provided. Insufficient knowledge and health awareness would prevent the use of appropriate healthcare resources and facilities, resulting in larger health gaps among vulnerable and susceptible populations as shown in this review and among pregnant women residing in the coastal areas in Sabah [14,17].

The WHO has developed the Sustainable Development Goals (SDGs) for health to ensure healthy lives and to promote the well-being of all individuals. The SDG are also used to achieve an improved level and distribution of universal health coverage (UHC) worldwide. The association between health and environment has been highlighted as part of the 2030 goals to substantially reduce the environment-related health problems [44]. It is significantly important to understand and investigate the association between the coastal ecosystems and human health to ensure that the health needs of the coastal communities are addressed accordingly. As Malaysia is progressing towards UHC, healthcare delivery and health education for individuals living in remote and rural parts such as in the coastal areas in Sabah should be strengthened to minimise health inequalities.

To date, this is the first systematic review conducted on the health status of the coastal communities in Sabah, Malaysia. This review provides substantial information to reflect on the healthcare burdens that are observed in the coastal areas in Sabah. The results from this study were also consistent with the findings from the previous literature that described explicitly on the significant health burdens experienced by individuals residing in the coastal communities, particularly in developing countries [1,2,3]. Nevertheless, the exclusion of non-full article journals, such as the national report and regional surveillance, might limit our review. We might have missed the prevalence and the information from these articles despite their interesting and important findings. Additionally, this review mainly focused on the health problems that were published in the journals. Although Sabah has a high prevalence of chronic non-communicable diseases [43], this systematic review could not elucidate these health problems probably due to the lack of studies conducted on the coastal communities. Therefore, a comprehensive, community study focusing on these problems with in-depth interviews and focus group discussion with relevant authorities could complement the objectives of investigating and understanding the health status of the individuals living in the coastal communities in Sabah.

## 5. Conclusions

In conclusion, this review provides an extensive review on the health status of the coastal community in Sabah, Malaysia. It is shown that the coastal communities are experiencing various health problems and health disparities still exist in the community. Findings from this review can be used by policy makers to plan for strategic measures that need to be taken in order to reduce the health gaps of the coastal community in Sabah, Malaysia. Appropriate and specific strategies to improve health issues are vital to improve overall health status and quality of life of the community. It is also important to address issues to enhance better health and wellbeing of the coastal communities in Sabah. Elucidation of the health status of the community will support more effective public health strategies and facilitate policy makers to identify possible intervention programs that are suitable to minimise the health challenges that the coastal community are experiencing.

## Figures and Tables

**Figure 1 ijerph-17-01533-f001:**
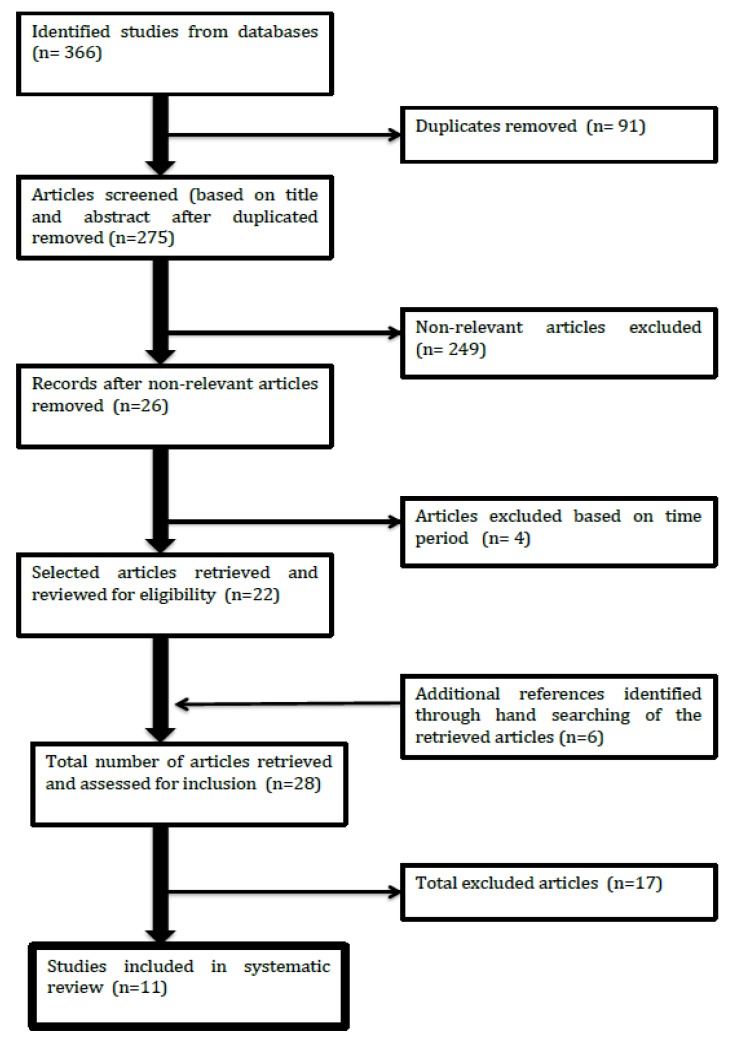
Flow of study.

**Table 1 ijerph-17-01533-t001:** Search strategy and keywords.

Health Status	Coastal	Community	Sabah
Health*‘health status’‘health-status’‘health impact*’‘health-impact*’‘health implication*’‘health burden*’‘health-burden*’‘health-related’‘health-related outcome*’‘health related outcome*’‘health outcome*’Illness*‘related illness*’Disease*‘disease status’‘disease-status’‘disease impact*’‘disease-impact*’‘disease implication*’‘disease burden*’‘disease-burden*’‘disease-related’‘disease-related outcome*’‘disease related outcome*’‘disease outcome*’‘quality of life’‘quality adjusted life years’‘quality-adjusted life years’‘health related quality of life’‘health-related quality of life’‘value of life’‘morbid*’‘mortalit*’‘death’‘life expectanc*’‘life-expectanc*’‘surviv*’‘healthcare utili*’‘healthcare-utili*’‘healthcare admission*’‘healthcare-admission*’‘healthcare visit*’‘healthcare-visit*’‘healthcare us*’‘healthcare-us*’	‘coast*’‘fish*’‘beach*’‘gulf*’‘bay*’‘sea*’‘ocean*’	‘communit*’‘societ*’‘people’‘resident*’‘popula*’	‘Sabah’‘borneo’‘north borneo’‘north-borneo’‘east Malaysia’‘east-Malaysia’‘Tun Mustapha park’‘Tun Mustapha marine park’‘Kudat’‘Pitas’‘Kota Marudu’‘Sabah park*’‘Sabah marine park*’
‘health’ [MeSH]‘health status disparities’ [MeSH]‘health care surveys’ [MeSH]‘health status indicators’ [MeSH]‘health status’ [MeSH]‘health impact assessment’ [MeSH]‘outcome assessment (healthcare)’ [MeSH]‘public health’ [MeSH]‘public health surveillance’ [MeSH]‘rural health’ [MeSH]‘quality of life’ [MeSH]‘quality-adjusted life years’ [MeSH]‘value of life’ [MeSH]‘disease*’ [MeSH]‘morbidity’ [MeSH]‘mortality’ [MeSH]‘survival rate’ [MeSH]‘facilities and services utilisation’ [MeSH]

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
