# Peer review of "A Review of Published Literature Regarding Health Issues of Coastal Communities in Sabah, Malaysia"

_ijerph, 2020, doi:10.3390/ijerph17051533_

Round 1

Reviewer 1 Report

In my opinion, the paper is well written and contributes to the existing knowledge. I could not find any logical errors in the presentation and the approaches used.

In addition, English should be checked, as there are some incorrect phrases (e.g.: In contrary; rely instead of relied, etc). The flow of study is easy to follow. Were PubMed TAGS used near keywords? Does "full article journal" mean available full text? Were brief reports included in the articles?

Author Response

We thank the reviewers for their comments, which have helped us to improve the manuscript. Our responses to the reviewers’ comments are as follows:

Reviewer 1:

English and grammar:

Response: We have re-send the manuscript to a professional proofreading service. We corrected all the language errors and we have improved style, clarity and sentence structure.

Were PubMed TAGS used near keywords?

Response: In this study, we used MeSH Subheadings and MeSH terms along with the keywords.

Does "full article journal" mean available full text? Were brief reports included in the articles?

Response: Full article journal is defines as published full reports of data from research that include details of study methodology and research findings, which include both descriptive study and analytical study. Descriptive study describes mainly the frequency, natural history and pattern/trend of disease. Descriptive study includes case series and case report. On the other hand analytical study is use to test one or more specific hypotheses of exposures and outcomes of interest. Analytical studies are designed to examine etiology and causal associations. Types of analytical studies are cross sectional, case-control, cohort (retrospective and prospective) and ecological.  In this systematic review, both descriptive and analytical studies were included. Otherwise, chapter in books, thesis, clinical registries and database and proceedings were excluded.

Reviewer 2 Report

The paper is a useful and informative piece of work. The only question I have about the paper was What exactly was the intent?  A "health-status assessment" of a community or region usually begins with a set of parameters or indicators which measure normal background "health" and "wellness".  In this case, the technique for deciding which parameters to use apparently consisted of a search of literature published in the last 10 years about health issues of concerns. 

It seems that the selection of the relevant criteria to evaluate the community was being left to the discretion of the authors of the selected papers, and the editorial staff of the journals in which the articles were published.  Typically, authors will write on a topic of interest, perhaps one that has been missed in the past, or a topic that is of particular interest to the authors. Frequently the novel or newly-emerging topics are of greatest priority to authors and editors alike.

In this case, the "assessment" is a selection of 4 topics: Malaria, seafood toxicosis, TB, and antenatal issues. I have no doubt these are genuine and relevant issues, but I wonder what happened to the USUAL indicators of population health.  Where are the infant mortality rate, maternal mortality rate, miscarriage rate, spontaneous abortion rate, proportionate mortality rate, age and sex specific mortality rates, and so on?  These are the background indicators used globally to assess the health status of any community.   Once these are established, emerging issues for that region can be "flagged" for special attention.  I believe the topics selected for this article are the "flags" rather than the essential indicators. 

I can understand if no previous monitoring of these communities was done, and this is the first attempt to assess them. But I find this hard to believe given the history of Malaysia and it's previous existence as Malaya. The article is titled "A Systematic Review of Health Status of Coastal Community in Sabeh, Malaysia". The method of searching the literature could be described as "systematic", BUT as far as a valid assessment of the community's health status, the work does not align with what is normally considered to be such an assessment.  

I suggest that a more appropriate title for the article might reflect "Selected issues affecting the health ..." or "Priority issues affecting health status of ...." etc.          

I will also mention the issue of tuberculosis as presented.  Pulmonary TB (person-to-person) and non-pulmonary TB (e.g. food and unpasteurized milk) have different sources and different ways they can be controlled. The use of haemoptysis suggests pulmonary TB but could also involve trachea, stomach, etc.   Can this be made clear, and if it's only pulmonary TB, is there any indication that non-pulmonary TB is not a problem? DRTB/ MDRTB/ XDRTB are extremely important issues these days.  Was drug resistance recorded? Measured?

Author Response

We thank the reviewers for their comments, which have helped us to improve the manuscript. Our responses to the reviewers’ comments are as follows:

Reviewer 2:

What exactly was the intent? 

Response: Thank you for bringing up this issue. Actually this systematic review is one of the components of a big research project that look at health status, health problems, health gaps, health needs, health utilization and quality of life among coastal communities in Sabah. Mainly, the big research project was planned to understand the health problems in the studied coastal area and to recommend suitable intervention programs to improve the overall health status of the costal community in future. This is because, many studies elsewhere shown that the coastal communities are facing with various health issues such as communicable diseases, non-communicable diseases, mental health problems and nutritional deficiencies. Nevertheless, researches focusing on these issues among coastal community in Sabah, Malaysia are lacking.

Coastal areas in Sabah are possibly unique compared to other coastal areas in Peninsular Malaysia and Sarawak. Many of the coastal zones in Sabah are isolated and remote with lack of accessibility. Even though public healthcare facilities are available in the coastal areas of Sabah, the facilities provided are limited and might not reach those who are needed due to many reasons such as poverty, poor health awareness and insufficient infrastructure to access the health services. On top of that, coastal areas in Sabah are experiencing the highest coastal erosion, which affects the livelihoods and health of the community. Sabah has the highest poverty rate in Malaysia and several coastal areas in Sabah such as Kudat, Kota Marudu, Pitas, Kunak and Semporna are reported to have poverty rates of near to 50%. Besides that, the number of stateless and undocumented people is also high in Sabah’s coastal areas. Because of these factors, the coastal community are exposed to many health challenges, which could result in poor health status.

Various sub-studies were planned to achieve the general objective. For example, in the first part of this big research project, a comprehensive systematic review of available literatures on health issues of coastal community in Sabah was conducted to get an overview of health problems and health gap among the coastal community.

Second part will include health surveys on general health, healthcare utilisation and quality of life among the community. While in the third part of this big research, in-depth interviews and focus group discussions with relevant authorities particularly those from Ministry of Health who are dealing with coastal community at Sabah will be conducted. At the same time, data collection from national registries and annual report will be collected accordingly.

We agreed with the reviewer about the health-status assessment with a set of parameters or indicators, which measure normal background "health" and "wellness” which can be gathered from national surveillance and report. However, as many of the coastal communities are stateless and poor, they don’t usually go to hospitals to receive treatment and therefore the surveillance systems and epidemiology report might not reflect the health of stateless and poor coastal community. Even though available published literatures are limited, those studies were conducted at community level and they might reflect and include people who are stateless and those who did not seek medical attention at public or private healthcare facilities.  

In Malaysia, health surveillance relies on information collected by healthcare facilities when patients come to seek treatment. This approach is limited in its ability to detect public health events and the occurrence of disease in populations that do not seek treatment at healthcare facilities or that experience barriers to treatment. This may be the case, for example, in hard-to-reach areas (remote communitiy, areas where the population relies highly on traditional healers or alternative treatments, or in populations with stigmatising illnesses such as HIV infections.

In this case, the "assessment" is a selection of 4 topics: Malaria, seafood toxicosis, TB, and antenatal issues:

Response: All the topics were from available published literatures. We only found these topics and group them accordingly. It indicates that the number of studies conducted in the coastal community, are limited. In Sabah epidemiology report, other than the four topics mentioned, there were also cases of food borne disease, vaccine preventable disease, zoonotic disease, sexual transmitted disease and others that were only captured among patients in care.

I wonder what happened to the USUAL indicators of population health.  Where are the infant mortality rate, maternal mortality rate, miscarriage rate, spontaneous abortion rate, proportionate mortality rate, age and sex specific mortality rates, and so on?  These are the background indicators used globally to assess the health status of any community.

Response: We are totally agreed with the reviewer. Those are the real and commonly used indicators. For this project, we planned to get data and the indicators during the in-depth interviews and focus group discussions with relevant authorities as mentioned above. We hope that this systematic review could compliment data from surveillance and registries as majority of the published study were conducted at community level that included patients who were not in care or patient who did not seek medical attention at any healthcare facilities.

Apart from that, as the big project will also include community survey, on top of the qualitative study (through in-depth interviews and focus group discussions), we hope that this systematic review will provide an overview on the general health of the coastal community before the other components of the study will be conducted.

We hope that with these three components or sub-studies, which include (systematic review, community survey and qualitative study) in this project, all of the information obtained can be assimilated and integrated to enhance our understanding of the health issues among the coastal community and in turn, we could plan and provide appropriate intervention to improve the health and wellbeing of the community.

I suggest that a more appropriate title for the article might reflect "Selected issues affecting the health ..." or "Priority issues affecting health status of ...." etc.          

Response: We are totally agreed with the reviewer. We understand that the term that we used before “health status” might be misleading and inappropriate.  We would like to change our title to: A review of published literature regarding health issues of coastal communities in Sabah, Malaysia. We hope that this new title could represent our manuscript better.

Can this be made clear, and if it's only pulmonary TB, is there any indication that non-pulmonary TB is not a problem? DRTB/ MDRTB/ XDRTB are extremely important issues these days.  Was drug resistance recorded? Measured?

Response: In Malaysia, national data shown that Sabah contributed the highest number of TB cases. Immigrants were the main contributing factors for TB cases in Sabah. This scenario was contradicted to the TB cases in other states of Malaysia, which was due to intravenous drug users and commonly occurred among HIV patients. In Sabah particularly, the main source of the disease is pulmonary TB while non-pulmonary TB through food, non-pasteurised milk and others is not a problem. Most of TB patients in Sabah contracted the disease through air-borne transmission from infected immigrants or foreigners. Based on national report on TB in Malaysia, HIV and hepatitis C were not the contributing factors for TB in Sabah and the most important factor is infected immigrants.

Reviewer 3 Report

Tables are not clear and hard to understand (formatting and information contained inside). It is not clear why this systematic review is being conducted in the first place. It is also not clear the context for such a review and what region this is coming (Malaysia) from and why it is important globally, as this is an international journal. Grammar and English is difficult to read. For a systematic review, this seems like too few references. 

Most importantly, what is the purpose of this study? A systematic review of health status? Wouldn't surveillance systems provide this information better than a review of scientific articles? 

Author Response

We thank the reviewers for their comments, which have helped us to improve the manuscript. Our responses to the reviewers’ comments are as follows:

Reviewer 3:

Tables are not clear and hard to understand (formatting and information contained inside)

Response: We have re-do the tables appropriately. We corrected all the language errors, formatting and we have improved style, clarity and sentence structure. We hope the tables are clear and easy to understand.

It is not clear why this systematic review is being conducted in the first place. It is also not clear the context for such a review and what region this is coming (Malaysia) from and why it is important globally. Most importantly, what is the purpose of this study? Wouldn't surveillance systems provide this information better than a review of scientific articles? 

Response: Thank you for bringing up this issue. Actually this systematic review is one of the components of a big research project that look at health status, health problems, health gaps, health needs, health utilization and quality of life among coastal communities in Malaysia. The big research project was planned to understand the health problems in the studied coastal area and to recommend suitable intervention programs to improve the overall health status of the costal community. This is because, many studies elsewhere shown that the coastal communities are facing with various health issues such as communicable diseases, non-communicable diseases, mental health problems and nutritional deficiencies. Nevertheless, researches focusing on these issues among coastal community in Malaysia are lacking.

Various sub-studies were planned to achieve the general objective. For example, in the first part of this big research project, a comprehensive systematic review of available literatures on health issues of coastal community in Sabah was conducted to get an overview of health problems and health gap among the coastal community.

Second part will include health surveys on general health, healthcare utilisation and quality of life among the community. While in the third part of this big research, in-depth interviews and focus group discussions with relevant authorities particularly those from Ministry of Health who are dealing with coastal community at Sabah will be conducted. At the same time, data collection from national registries and annual report will be collected accordingly.

Coastal areas in Sabah were chosen because they are possibly unique compared to other coastal areas in Peninsular Malaysia and Sarawak. Many of the coastal zones in Sabah are isolated and remote with lack of accessibility. Even though public healthcare facilities are available in the coastal areas of Sabah, the facilities provided are limited and might not reach those who are needed due to many reasons such as poverty, poor health awareness and insufficient infrastructure to access the health services. On top of that, coastal areas in Sabah are experiencing the highest coastal erosion, which affects the livelihoods and health of the community. Sabah has the highest poverty rate in Malaysia and several coastal areas in Sabah such as Kudat, Kota Marudu, Pitas, Kunak and Semporna are reported to have poverty rates of near to 50%. Besides that, the number of stateless and undocumented people is also high in Sabah’s coastal areas. Because of these factors, the coastal community are exposed to many health challenges, which could result in poor health status.

This study was aimed primarily at improving the health,well being and quality of life of the coastal communities. One of the rationale behind this study is that, Malaysia has achieved universal health coverage (UHC) status and has made remarkable progress in improving health outcomes over the past seven decades. In Malaysia, it is the right for all the people to get access to health without facing difficulties and extreme poverty due to health care costs. Despite the UHC acvievement, there are some communities that still have trouble accessing health facilities, especially among the poor and rural communities including the cosatal communities. It is more apparent among coastal communities in Sabah because of tall the factors mentioned above.  Findings from this SR can be used to understand the health problems and issues encountered by the community and can be applied to the coastal communities in other Sout-east Asia countries in view of the similar geographical and cultural structure with Malaysia.

Grammar and English is difficult to read.

Response: We have re-send the manuscript to a professional proofreading service. We corrected all the language errors and we have improved style, clarity and sentence structure.

For a systematic review, this seems like too few references. 

Response: Although the number of studies available are limited, this clearly indicates that there is very little research conducted for these communities. Therefore, we hope that this systematic review togeter with other sub-studies as mentioned above  will help to increase the understanding and attention to coastal communities

Round 2

Reviewer 3 Report

I believe this article has been edited and re-framed well. The issue still remains, how significant is this systematic review of a particular sub-populations general health as it relates to the international health scene? 

Author Response

We thank the reviewers for their comments, which have helped us to improve the manuscript. Our response to the reviewers’ comment is as follows:

I believe this article has been edited and re-framed well. The issue still remains, how significant is this systematic review of a particular sub-populations general health as it relates to the international health scene?

Thank you for bringing up this issue. As far as we are concerned; this systematic review is the first systematic review on health status of coastal communities in South East Asia, which could contribute to the body of knowledge. Many previous studies were conducted among communities in developed countries such as in Europe and United States. While studies in UK found that people living at coastal areas have good health status and better quality of life compared to non-coastal people, due to opportunities for physical activity and psychological well-being, coastal people in poor and developing countries are facing with numerous health issues due to uncontrolled climate changes, overcrowded, lacking of clean water supply, improper waste disposal and many others. Those factors expose them to many diseases. Information obtain from this systematic review could helps us to understand the health issues so that relevant actions can be taken to improve their health status and well being.

Coastal communities are both uniquely valuable and vulnerable. While the coastal areas are important for global trade, tourism, agriculture, aquaculture, and fishing, the settlements are also vulnerable to climate changes such as sea level rise and intense coastal storms, which might affect their health status and quality of life. We believe that, proper actions on managing the coastal communities require an interdisciplinary approach that has a solid foundation in health, natural science, social science, and policy, nationally and internationally. Therefore, information obtained from this study can provide and strengthen awareness on coastal communities particularly in South East Asia, which have common geographical structure, culture and lifestyle and similir health care system. We learned and understand that there is no such thing as a one-size fits all approach in working with vulnerable coastal communities. Therefore, contest-specific strategies are fundamental and this study can be a stepping-stone to help other developing countries to raise awareness about the health problems facing by the coastal communities.

Through this project, we also realized that the people who are the most vulnerable at the coastal areas are often the poorest. It is essential that we act upon what we know in order to build resilience in the poorest communities. A large portion of the world’s poor coastal people lives in Asia and the Pacific. Therefore it is important to understand the welfare of coastal communities in order to build resilience and institute sustainable practices so that vulnerable people can make a living while also preserving the environment and the resources. Poor lifestyle habits such as smoking, poor diet, and physical inactivity are more prevalent in poor people, such as the coastal communities. The use of preventive health services such as regular medical checkups or consult minor health problems with a physician is a rare practice in the lower socioeconomic group and it is worsened with the lack of health care facilities in that area. Under universal health coverage (UHC), the full spectrum of quality and efficient health services is fundamental. Even though many countries are now have successfully achieved UHC target including Malaysia, the health issues among poor, remote and vulnerable coastal communities remain challenging. Therefore, this systematic review accessed the published health issues of people of coastal areas, to understand the vulnerability of populations, and potential adaptations and possible health interventions that could be beneficial to the community in order to improve the UHC. Findings generated from this systematic review could also help other countries to improve healthcare access and health status of the coastal communities especially among the poor